# Antifungal Activity and Effect of Plant-Associated Bacteria on Phenolic Synthesis of *Quercus robur* L.

**DOI:** 10.3390/plants12061352

**Published:** 2023-03-17

**Authors:** Svitlana Bilous, Artur Likhanov, Vira Boroday, Yurii Marchuk, Liubov Zelena, Oleksandr Subin, Andrii Bilous

**Affiliations:** 1Education and Research Institute of Forestry and Landscape-Park Management, National University of Life and Environmental Sciences of Ukraine, 03041 Kyiv, Ukrainemarchuk_iuriy@nubip.edu.ua (Y.M.);; 2Institute for Evolutionary Ecology NAS of Ukraine, 37 Lebedeva Str., 03143 Kiev, Ukraine; 3Forestry Department, Weihenstephan-Triesdorf University of Applied Sciences, Germany, Hans-Carl-von-Carlowitz-Platz 3, 85354 Freising, Germany; 4Danylo Zabolotny Institute of Microbiology and Virology National Academy of Sciences of Ukraine, 154 Zabolotnogo Str., 03143 Kyiv, Ukraine; 5State Enterprise “State Centre of Agricultural Products Certification and Examination”, Janusha Korchaka Str. 9/12, 03143 Kyiv, Ukraine

**Keywords:** endophytic bacteria, oak, antagonistic activity, polyphenols

## Abstract

Europe’s forests, particularly in Ukraine, are highly vulnerable to climate change. The maintenance and improvement of forest health are high-priority issues, and various stakeholders have shown an interest in understanding and utilizing ecological interactions between trees and their associated microorganisms. Endophyte microbes can influence the health of trees either by directly interacting with the damaging agents or modulating host responses to infection. In the framework of this work, ten morphotypes of endophytic bacteria from the tissues of unripe acorns of *Quercus robur* L. were isolated. Based on the results of the sequenced 16S rRNA genes, four species of endophytic bacteria were identified: *Bacillus amyloliquefaciens*, *Bacillus subtilis*, *Delftia acidovorans*, and *Lelliottia amnigena*. Determining the activity of pectolytic enzymes showed that the isolates *B. subtilis* and *B. amyloliquefaciens* could not cause maceration of plant tissues. Screening for these isolates revealed their fungistatic effect against phytopathogenic micromycetes, namely *Fusarium tricinctum*, *Botrytis cinerea*, and *Sclerotinia sclerotiorum*. Inoculation of *B. subtilis*, *B. amyloliquefaciens*, and their complex in oak leaves, in contrast to phytopathogenic bacteria, contributed to the complete restoration of the epidermis at the sites of damage. The phytopathogenic bacteria *Pectobacterium* and *Pseudomonas* caused a 2.0 and 2.2 times increase in polyphenol concentration in the plants, respectively, while the ratio of antioxidant activity to total phenolic content decreased. Inoculation of *Bacillus amyloliquefaciens* and *Bacillus subtilis* isolates into oak leaf tissue were accompanied by a decrease in the total pool of phenolic compounds. The ratio of antioxidant activity to total phenolic content increased. This indicates a qualitative improvement in the overall balance of the oak leaf antioxidant system induced by potential PGPB. Thus, endophytic bacteria of the genus *Bacillus* isolated from the internal tissues of unripe oak acorns have the ability of growth biocontrol and spread of phytopathogens, indicating their promise for use as biopesticides.

## 1. Introduction

Plants are constantly exposed to stress factors of biotic and abiotic nature. As a result of climate change, changes in zonal vegetation types and a significant increase in areas with unfavourable conditions for the growth of woody plants are expected by the end of the XXI century. Under such conditions, a substantial decrease in the productivity of forest-forming species, a weakening of their resistance to pests and diseases, and a gradual loss of reproductive capacity and the possibility of natural regeneration of forests is predicted [1,2,3]. Due to global challenges, the development of biological compensatory mechanisms that support the sustainability of forest ecosystems is particularly relevant.

One of the components of the plant resistance system under extreme conditions consists of endophytic microorganisms, which together with epiphytes form associated microbiomes [4]. Recently, much data have been obtained confirming positive effects of plant-associated bacteria on plants [5,6,7,8]. Their unique role in maintaining the viability of the plant organism is demonstrated by the fact that endophytes are found in the tissues of almost all plant organs: roots, shoots, flowers, fruits, and seeds [6,7,8,9]. However, the interaction mechanisms between plants and endophytic microorganisms are quite complex and depend on many factors [10,11].

Nowadays, special attention has been paid to studies of taxonomic groups of endophytes, potential ways of colonization of plant tissues and organs [12,13], physiological processes responsible for the formation of the acquired resistance of plants [14,15,16], and plant-microbial interactions and expression of related genes [17,18,19,20]. Some species of endophytic bacteria are capable of fixing atmospheric nitrogen [21,22,23,24]. They synthesize biologically active substances and plant growth regulators [25,26,27,28,29]. Endophytes participate in protection of plants from infections [30] and show antagonistic activity against pathogens [31,32]. It is known that bacteria of the genera *Rhizobium*, *Bacillus*, *Pseudomonas*, *Pantoea*, *Paenibacillus*, *Burkholderia*, *Achromobacter*, *Azospirillum*, *Microbacterium*, *Methylobacterium*, *Variovorax*, and *Enterobacter* can increase plants’ resistance to abiotic stresses [33,34].

Among the microorganisms associated with plants, a specific group called Plant Growth Promotion Bacteria (PGPB) deserves particular attention. These bacteria occupy ecological niches in the phyllosphere, rhizosphere, and endosphere of the plant organism and they are in symbiotic interaction with them. The plant endosphere, compared to other groups of PGPB, represents a specific and insufficiently studied niche where only about 2% of the list of plant species inhabited by endophytic microorganisms has been studied [17,35,36,37].

In the context of conservation and restoration of forests, it is extremely important to study the importance of endophytic bacteria for unifier plants that can significantly transform the environment. One of the main forest-forming species in forest ecosystems of Europe is the common oak (*Quercus robur* L.). The oak phyllosphere is extremely diverse [38,39]. To maintain a stable symbiosis, endophytes produce biologically active substances. They increase plant resistance to abiotic and biotic stresses [40,41,42,43]. Endophytic bacteria often have long-term ecological interactions with host plants, including symbiosis, mutualism, and commensalism (Table 1).

Isolated from the endo- and rhizosphere of the root system of *Quercus* spp. bacteria of the genera *Luteibacter*, *Pseudomonas*, and *Arthrobacter* showed that in the complex of cell-associated hydrolytic enzymes [44], the vast majority of their total activity belongs to strains of the genus *Luteibacter*. Cellulolytic and hemicellulolytic enzymes are quite common among strains of the genera *Luteibacter* and *Pseudomonas* and are rarely found in *Arthrobacter* [44]. Cell-associated enzymes of bacteria of the root system of *Quercus* spp. are an essential factor in the transformation of organic matter [45], affecting the cycles of C, N, and P, and significantly impact the functionality of forest ecosystems [46,47,48,49]. Endophytes show chemotaxis in relation to root exudates of host plants [50,51,52,53]. This is particularly important for middle-aged plants and perennial trees, which have a stabilizing effect on forest ecosystems as a whole [54,55]. Study of the features of *Quercus petraea* (Matt.) Liebl colonization by endophytes and their further influence on the plants’ growth and development in alpine forests was carried out [56].

**Table 1 plants-12-01352-t001:** Study of the endophytic mycobiota of *Quercus* spp.

Region	Species	Ecological Niche	Authors
Central, Western, Southern, and Northern Europe	*Q. petraea*	endophytes of leaves, shoots and branches of different ages	Halmschlager et al., 1993, Fort et al., 2021 [17]
	*Q. robur*		Griffith et Boddy, 1990, Petrini et Fisher, 1990, Kowalski et Kehr 1996, Gennaro et al., 2003, Ragazzi et al., 2003, Gonthier et al., 2006, Agostinelli et al., 2018, Matule, 2018 [54,55,56,57,58,59]
	*Quercus ilex*	the leaves and twigs	Fisher et al., 1994, Collado et al., 1996, 1999 [60,61]
	*Q. faginea*	the leaves and twigs	Collado et al., 1996, 1999 [60]
	*Q. cerris*, *Q. pubescens*	the leaves and twigs	Ragazzi et al., 2001, 2003, Gennaro et al., 2003, Moricca et al., 2012 [59,62]
	*Q. suber*	young and old twigs, branches, and woody tissues	Linaldeddu et al., 2011, Costa et al., 2019 [63,64]
Asia	*Q. macranthera*, *Q. brantii*	twigs, branches	Ghasemi-Esfahlan et al., 2019 [65]

Relationships between nitrogen content, activity of the stomatal apparatus, and the degree of leaf damage by herbivores and seasonal dynamics of endophytic bacterial complexes were revealed [66].

In recent years, there has been a growing scientific interest in the biodiversity and functions of endophytic bacteria [67], as well as the prospects for their practical use [68,69]. Thus, in the rhizosphere of forest trees affected by fires, more than 21% of the total number of microorganisms is represented by the genus *Arthrobacter*. Most of its strains showed the ability to decompose organic polymers in vitro and stimulate plant growth. On this basis, representatives of the genus *Arthrobacter* were considered as PGPRs that can contribute to forest regeneration after fires [70]. The possibility of using endophytic bacteria as the basis of biofertilizers is being actively investigated [71]. Associated species of endosymbiont fungi isolated from oak tissues are being studied as biocontrol agents to suppress oak pathogens [72].

Endophytes that colonize seeds are transmitted to the next generations through vertical transmission. This provides essential features of plant growth determined by the genomes of both microorganisms and plants [73,74]. The endophytic microbiota of seeds consists of a limited range of species and evolves through co-evolution with host plant species [75]. Since the reproductive organs of plants are the most protected from pathogens (pathogenic external microbiota), endophytic bacteria and fungi that colonize them can overcome tissue barriers and, without causing damage to seeds or diaspores, can be transmitted from the mother plant through generations. The genotype of the donor plant and the place from which natural endophytes are isolated are significant [35,76,77].

Here the aim was to isolate and identify endophytic bacteria from the tissues of immature *Quercus robur* L. acorns and to study their biological activity about pathogenic micromycetes and seedlings of *Q. robur*.

## 2. Results and Discussion

### 2.1. Determination of Pectolytic Enzymes Activity

Ten samples of endophytic bacteria were isolated from the tissues of the embryos of unripe acorns of common oak, among which there were dominant morphotypes. The absence of microorganisms from the surface washes of sterilized acorns confirms that the isolated bacteria were endophytic.

Determining the activity of pectolytic enzymes causing the maceration of plant tissues showed that the isolates, conditionally labelled as Q2 and Q7, could not macerate. In isolates Q5 and Q6, pectolytic activity was weakly expressed. The remaining isolates showed medium and high pectinase activity and were not used in further studies.

For the identification and study of ecophysiological properties, four typical representatives of the dominant morphotypes were chosen: bacterial isolates Q2, Q5, Q6, and Q7. They were isolated in pure culture, determined by Gram and sequenced for species identification. Microscopic studies showed that the cells of Q2 and Q7 cultures were Gram-positive, while Q5 and Q6 were Gram-negative bacteria with different cell morphologies. For more accurate identification of selected isolates, sequencing of 16S rRNA gene fragments was performed.

### 2.2. 16S rRNA Gene Sequencing

The fragments of 1454 nucleotides (strain BAQ7-PSTQR-0920) as well as 564 and 511 nucleotides (strain BSQ2-PSTQR-0920) were obtained as a result of 16S rRNA gene sequencing. The BLAST analysis of these sequences and those deposited in GenBank revealed 99,86% similarity between strain BAQ7-PSTQR-0920 and *Bacillus amyloliquefaciens* strains while the highest similarity, 99,64%, was observed between strain BSQ2-PSTQR-0920 and *B. subtilis* strains. To confirm obtained results and to analyze phylogenetic relationships between new strains and various representatives of *Bacillus* genus, the dendrograms of genetic similarities between bacilli were constructed (Figure 1 and Figure 2). It was shown that the strains BAQ7-PSTQR-0920 and BSQ2-PSTQR-0920 were included in the groups of *B. amyloliquefaciens* and various strains of *B. subtilis*, respectively. The 16S rRNA gene nucleotide sequences were submitted to GenBank database with the accession numbers MW282171.1 (*B. amyloliquefaciens* BAQ7-PSTQR-0920), MW282172.1, and MW282173.1 (*B. subtilis* BSQ2-PSTQR-0920).

The identification of strains Q6—*Delftia acidovorans* (DQ6-BEQR-0929) and Q5 *Lelliottia amnigena* (LQ5-BEQR-0928) was performed in the same way. The 16S rDNA sequences of the strains LQ5-BEQR-0928 (1061 nucleotides) and DQ6-BEQR-0929 (801 nucleotides) were obtained and compared with those in GenBank using blastn. The sequences’ identity of 99,47% was revealed between the strain LQ5-BEQR-0928 and the *L. amnigena* and *L. jeotgali* species; additionally, the similarity between the new strain and *L. nimipressuralis* was 98,94%. The comparative analysis using blastn showed 99,85% similarity between the strain DQ6-BEQR-0929 and *D. acidovorans* strains.

The phylogenetic dendrograms for isolated strains were constructed to clarify species identification and define the genetic relationships with various close relative species (Figure 3). As is shown in Figure 3a, the strain *Delftia* sp. DQ6-BEQR-0929 and the *D. acidovorans* type strain formed the group that is separated from the others (the bootstrap value is 90). It confirms that DQ6-BEQR-0929 belongs to *D. acidovorans*. Figure 3b shows that the strain LQ5-BEQR-0928 together with various strains of *L. amnigena* and *L. nimipressuralis* species are combined in a group to which the group of *L.jeotgali* strains is joined. It suggests high similarity between these species and supposes using additional markers (for example, morphological, biochemical, or genetic markers) to identify bacterial species. Thus, on the basis of a set of morphological, cultural, and genetic characteristics, the strain LQ5-BEQR-0928 was defined as *L. amnigena*. The 16S rRNA gene sequences were submitted to the GenBank with accession numbers: OK560290.1 and OK560292.1 (*D. acidovorans* strain DQ6-BEQR-0929) and OK560291.1 and OK560293.1 (*L. amnigena* strain LQ5-BEQR-0928).

The obtained results confirmed the report that bacteria of the genera *Bacillus* and *Delftia* were endophytes of oak (*Q. robur*). These and other isolates synthesize indole-3-acetic acid, respectively, promoting plant growth (PGPBs) [78,79]. Bacteria *B. amyloliquefaciens* and *B. subtilis* (phylum *Bacillota*) and *Lelliottia amnigena* and *Delftia acidovorans* (phylum *Pseudomonadota*) are capable of nitrogen fixation, mineralization/solubilization of phosphates, and formation of siderophores [79].

At the same time, we failed to isolate bacteria of the genus *Paenibacillus* from the internal tissues of unripe acorns of *Q. robur*, as reported by the other authors [80]. A possible explanation for their absence among the obtained isolates is the presence of a complex system of tissue and cellular barriers in the generative organs. It is obvious that not all microorganisms, especially phytopathogenic ones, can overcome them without destroying or damaging seed tissues. Therefore, plants’ reproductive organs can be considered a reservoir for storage and subsequent vertical transfer to the next generation of potentially useful microorganisms.

Thus, endophytes isolated from oak seeds increase the resistance of plants in nurseries against the damping off. This is one of the most common and dangerous diseases caused by pathogenic micromycetes *Fusarium* spp., *Alternaria* spp., *Rhizoctonia* spp., *Verticillium* spp., *Botrytis* spp., *Sclerotinia* spp. etc. [34]. Inoculation of *Pseudomonas denitrificans* bacteria into oak seedlings grown in containers after subsequent infection of the plants with oak wilt (*Raffaelea quercus-mongolicae*) reduced the number of diseased plants by half [18,79,81,82]. Therefore, the ability of endophytic bacteria to biocontrol the growth and spread of phytopathogens reveals the prospects of their use as biopesticides.

### 2.3. Antifungal Activity of Plant-Associated Bacteria

Screening of the antimicrobial activity of isolates of *B. subtilis* (BSQ2-PSTQR-0920) and *B. amyloliquefaciens* (BAQ7-PSTQR-0920) isolated from acorns revealed their fungistatic effect against pathogens, namely *Fusarium* spp., *Alternaria* spp., *Botrytis cinerea*, and *Sclerotinia sclerotiorum*. At the same time, the fungistatic activity of these endophytes differed somewhat (Table 2). The isolate BAQ7-PSTQR-0920 demonstrated the ability to significantly inhibit the growth of phytopathogenic micromycetes *F. tricinctum*, *B. cinerea*, and *S. sclerotiorum*. The bacterial culture BSQ2-PSTQR-0920 had less activity. The growth inhibition zone (GIZ) of the phytopathogens mycelium by Q7 exometabolites ranged from 4.0 to 13.2 mm. It was about three times higher than in the corresponding indicator in Q2.

The GIZ for this endophyte ranged from 1.3 to 5.1 mm, depending on the type of fungus. Other isolates were less active. Within five days, under the influence of exometabolites Q2 and Q7, the fungal colonies acquired an elliptical shape. The radii of *B. cinerea mycelia* (R_av_) close to the Q7 bacterial growth zone averaged 11.0–13.1 mm, while R_1_ (remote from the bacteria) averaged 22.2 mm. It should be mentioned that the transfer of the mycelium discs of *B. cinerea* to the nutrient medium was accompanied by the dispersion of spores. In the control variant, they formed numerous colonies that restrained mycelium growth from the edge of the discs (Figure A1a). However, in the presence of bacteria Q2 and Q7, only single colonies with slow growth were formed (Figure A1b,c).

Similarly, the radii (R_av_) of *S. sclerotiorum* were 13.8–16.6 versus 22.7 mm (Figure A1d–f), and those of *F. tricinctum* were 7.2–7.4 versus 18.2 mm (Figure A1g,i). The ratio of radii R_1_/R_av_, distant and close to bacteria, turned out to be a rather informative indicator of the antifungal activity of endophytes. Under the influence of Q7 metabolites, it was 1.5–1.9, and for Q2 it was 1.2–1.7. In the control samples, the shape of the mycelium was close to a regular circle (R_1_/R_av_ = 1.0–1.1). The revealed asymmetry of mycelial growth on nutrient media partially characterizes the interaction of endophytic bacteria with pathogenic fungi during their penetration into plant tissues and organs.

### 2.4. Reactions of Oak Seedlings to the Inoculation of Endophytic, Epiphytic and Phytopathogenic Bacteria

The effect of potential growth-stimulating bacteria on *Q. robur* seedlings was studied by inoculation into the leaf tissue due to minor damage to the integrity of the epidermis by abrasive material. The effect of the selected endophytic bacteria was compared with the effect of phytopathogenic and epiphytic bacteria. During field testing of bacteria on one-year seedlings of *Q. robur*, it was established that over time phytopathogenic bacteria *Pectobacterium* spp. and *Pseudomonas* spp. formed quite significant lesions on the leaves with characteristic depigmentation and necrotic zones (Figure 4b,c). It should be noted that as a result of injury to the surface of the leaves, traces of damage also remained in the control plants.

After inoculation with isolates Q2, Q7, and their complex (1:1), there were no signs of damage to the leaf plates in the inoculation zones. The fact that during the inoculation of bacteria the surface of the oak leaves was partially damaged by an abrasive material to ensure the penetration of bacterial cells into plant tissues deserves special attention. However, the isolate Q7 and a mixture of bacteria Q2 and Q7, in contrast to other types of bacteria and the control, not only did not affect the leaves but also contributed to the complete restoration of the epidermis in the places of damage.

One explanation for this effect could be the rapid and complete regeneration of damaged leaf tissue. This is possible due to the action of bioactive compounds, in particular phytohormones, that are synthesized by these endophytic bacteria species [78,79]. However, more research is needed to confirm this assumption.

Common oak seedlings reacted slightly differently to the inoculation of *Lelliottia amnigena* (Q5) and *Delftia acidovorans* (Q6) bacteria. Small areas of tissue damage were formed on the surface of the leaves. For isolate Q5, the lesion area did not exceed 1.3% of the total leaf area and did not differ from the control (at the *p* < 0.05 level). Under the influence of Q6 bacteria, fairly numerous but small lesions were formed on the surface of the leaves over time, the total area of which was significantly larger than the control (Figure 5).

However, it should be noted that these damages were local and did not affect the general condition of the seedlings. Zones of tissue damage were also detected in the inoculation zones of epiphytic Gram-positive bacteria EpQ1 and EpQ2, which were isolated from the surface of common oak leaves. The nature of the impact of epiphytes differed from endophytes by reducing the number of affected zones but increasing their area. As mentioned above, these bacteria have weak pectinase activity, which indicates their ability to destroy polysaccharides that provide intercellular communication. The excessive activity of this enzyme is dangerous for the plant. At the same time, it is obvious that the metabolism of bacteria depends on the environment, and under conditions of balanced interaction with the plant organism, the risks of damage to tissues and organs are significantly reduced.

Phenolic compounds are an important factor in regulating the activity of microorganisms in plants. Hydrolyzed and condensed tannins play a particularly important role in this. Common oak plants are extremely rich in hydrolyzed tannins, affecting their interaction with tolerant endophytes. An increased content of polyphenols usually inhibits bacterial growth, which is undesirable for plant-associated microorganisms. Accordingly, the plant-microbial system needs the co-adaptation of organisms, which creates a corridor of physiological comfort for potential symbionts. This is indicated by the results of determining the content of polyphenols in the leaves of oak seedlings after inoculation of potential PGPB, as well as pathogenic and conditionally pathogenic bacteria (Table 3).

Inoculation of bacteria Q2, Q7, and their sum was accompanied by a decrease in the number of total phenols in the leaves. The response of plants to isolate Q7 was the greatest. In plants inoculated with this strain, the number of phenolic compounds decreased by 15–20% compared to the control. In contrast to them, phytopathogenic *Pectobacterium* and *Pseudomonas* caused an increase in the concentration of polyphenols in seedlings by 2.0 and 2.2 times, respectively. Oxidative stress, which is usually accompanied by the invasion of phytopathogens, causes an increase in the total amount of phenolic antioxidants.

At the same time, it should be noted that the ratio of antioxidants to the number of total phenols in plants inoculated with potential PGPB and phytopathogens differed significantly.

Therefore, for isolate Q2, this indicator was 1.59; for Q7, it was 1.44; for the complex Q2 and Q7, it was 1.51. This ratio was significantly lower in the leaves of seedlings under the influence of phytopathogens: for *Pectobacterium*, the ratio was 1.10, and for *Pseudomonas* bacteria the ratio was 0.99. Consequently, an increase in phenolic compounds against the background of a significant increase in antioxidant potential indicates a high pool of reduced phenolic compounds, which can perform protective functions and provide higher resistance to plants against the background of a slowdown in phenylpropanoid synthesis, which takes away the energy resources of the plant organism. On the contrary, the body’s resistance to phytopathogens occurs against the background of increased phenolic synthesis.

In conditions of oxidative stress under the influence of oxides, phenolic compounds can be transformed into extremely active catechins, which are capable of damaging the cell membranes of pathogenic microorganisms. At the same time, due to high oxidative stress, a significant amount of radicals are formed in cells, which can damage cell membranes due to the peroxidation of lipids. Phenolic antioxidants neutralize possible negative consequences of excessive production of radicals.

However, there are different data on the antioxidant potential of plants under stress conditions. There are reports that PGPM (Plants Growth Promotion Microorganism) promotes antioxidant activity in response to drought stress [83,84,85,86]. Other studies have shown that PGPM inoculation correlates with decreased antioxidant activity. This fact is explained by the fact that PGPM have additional and little-studied mechanisms of stress reduction, which subsequently leads to a slowdown in ROS production and, as a result, to a decrease in antioxidants [87].

Analysis of the profiling results of phenolic complexes in the leaves of common oak seedlings by the method of principal components confirmed that the studied bacteria are grouped into separate classes according to the nature of their effect on the plant organism after inoculation. Therefore, the index of the ratio of the phenolic antioxidants activity to the total phenols (AA*i*/Ph*i*) can be an informative marker that allows distinguishing PGPB itself from other groups of endophytes in plant-microbial systems. Thus, for epiphytes, these indicators were close to phytopathogens: EpQ1 was 1.02 and EpQ2 was 0.96. For endophytes with pectinase activity (Q5 and Q6), they were 0.86 and 1.20, respectively. Based on this indicator, two species can be assigned to the PGPB class from the selected endophytes to some extent: the *B. subtilis* strain BSQ2-PSTQR-0920 and the *B. amyloliquefaciens* strain BAQ7-PSTQR-0920. Close to them is *D. acidovorans* DQ6-BEQR-0929. Analysis of the results of profiling of phenolic complexes in the leaves of common oak seedlings by the method of principal component analysis confirmed that the studied bacteria were grouped into separate classes according to the nature of their effect on the plant organism after inoculation. Potential PGPB in the coordinates of the components F1 and F2 (Figure 6a) and F1 and F3 (Figure 6b) were close to each other as well as to the control plants.

Since on the basis of a difference in the composition of the products of secondary synthesis endophytic bacteria create groups and are separated from pathogens and epiphytes, it is obvious that plants are able to recognize them and respond to them accordingly. Among the examined features (biochemical phenols) in oak leaves, the largest contribution to the total variance was made by the total pool of polyphenolic compounds and phenolic antioxidants. This confirms the key role of tannins, which make up the vast majority of common oak phenols, in the interaction of the plant with microorganisms. This is particularly confirmed by the ability of oak bark extract (*Quercus* sp.) to reduce the synthesis of acyl homoserine lactones (acyl-HSL), a signalling molecule responsible for bacterial intercellular communication known as quorum signalling (QS), through suppression of the QS-related genes expR/expI [88].

Besides tannins, indicators of the total content and number of individual flavonoids contained in oak leaves were quite labile to bacterial inoculation. Flavonoids with Rf ~0.83, Rf ~0.71, and 0.78 had the greatest contribution to the total variance of the second main component (F2 axis). The content of the latter two increased in plants that were inoculated with EpQ1 epiphytes and pectolytic bacteria. It is possible that the increase in the number of flavonoids in the extracts was related to the ability of pectinases to release them due to the hydrolysis of polysaccharides [88]. At the same time, flavonoids are able to interact with proteins and regulate their function. An increase in the concentration of flavonoids in plant tissues can affect the activity of exoenzymes of endophytes, pathogenic and conditionally pathogenic bacteria, and fungi.

Thus, the content of quercetin glycosides increased in the wood of common oak affected by brown rot. In the leaves of trees with signs of abnormal browning of the wood, the balance in the composition of flavonoids shifted towards an increase in the concentration of kaempferol glycosides [89]. It is known that the precursor of kaempferol is dihydrokaempferol. The latter is synthesized from naringenin by the enzyme flavanone 3-hydroxylase (F3’H). Precursors of quercetin are dihydroquercetin, which is synthesized from dihydrokaempferol under the action of flavonoid-hydroxylase (F3’H). The synthesis of eriodictyol, which is an alternative precursor of dihydroquercetin, also depends on F3’H. Therefore, both ways of dihydroquercetin synthesis, from which quercetin is further formed with the participation of flavonol synthase (FLS), depend on the work of F3’H [90]. Therefore, a decrease in the activity of this enzyme due to internal reasons or under the influence of endophytic microorganisms or other factors will lead to an increase in the proportion of kaempferol and its glycosides in plant tissues.

The ability of phytopathogenic bacteria to influence the metabolic profiles of common oak phenols is also confirmed by the fact that in the coordinate plane of the main components they united in separate pairs with epiphytes EpQ1 from *Pectobacterium* and EpQ2 from *Pseudomonas*. Pairs composed in this way can testify to the similarity of their effects on the plant, despite their phylogenetic distance. The results of PCA analysis showed that in secondary synthesis, the most sensitive products to EpQ2 and *Pseudomonas* strains are two phenolic compounds (Rf ~0.42 and Rf ~0.57). According to the position on the chromatogram and the characteristic blue fluorescence in the UV (365 nm), they are probably conjugates of phenolcarboxylic acids. The explanation of this rather unexpected affinity of bacteria of different genera is related to the peculiarities of the plant’s reaction to them. Bacterial strains of the genera *Pseudomonas* and *Bacillus* are known to be the most common groups inducing ISR [91,92]. At the same time, *Bacillus* species colonize oaks more easily than *Pseudomonas* species. Carroll G.C. in 1988 suggested that endophytes play an important role in inducing systemic resistance in trees. Because the life cycles of endophytes are much shorter than those of their host plants, bacteria evolve much faster. This creates prerequisites for a high selection of antagonists against pests and pathogens. The systemic resistance to diseases induced in plants by *Bacillus* spp. is made possible by increasing salicylic acid content and the gene and protein expression of proteinase inhibitor II (Pin2) and pathogen-resistant 1 (PR1) [93]. Therefore, one of the main directions of the practical application of the studied endophytes as a basis for biological preparations is the search and study of new types of endophytic microorganisms [34]. This gives reason to claim that the use of PGPE or complexes based on PGPB and PGPE is a promising and relevant direction in optimizing the technology of adaptation of woody plants grown in nursery conditions and adapted after in vitro culture [81].

## 3. Materials and Methods

### 3.1. Study Area

Centuries-old *Q. robur* trees (age about 200 years old) with signs of increased resistance to phytopathogens and pests were selected as donor plants [94]. The research was conducted in the educational and scientific laboratory of cell engineering and biotechnology and the laboratory of industrial biotechnology of the National University of Life and Environmental Sciences of Ukraine.

Isolates of endophytic microorganisms isolated from the tissues of immature oak acorns, cultures of phytopathogenic micromycetes, and pathogens of woody plants were used in the research.

### 3.2. Isolation, Cultivation and Studying of the Biological Activity of Endophytic Bacteria

#### 3.2.1. Isolation and Cultivation of Bacteria

The isolation of endophytic bacteria from the tissues of embryos of immature oak acorns [94] and epiphytic bacteria (EpQ1, EpQ2) from leaves was performed as reported in Borkar [95] and phytopathogenic bacteria research methods [96]. Bacteria were cultivated on the PDA medium (Potato Dextrose Agar). The bacteria were Gram-stained and used to study the morphological features of bacteria. Additionally, spore formations and the character of growth on solid and liquid media of the endophytic microorganisms were studied. Isolates of endophytic microorganisms were identified according to morphological and cultural properties according to generally recognized methods in bacteriology [96]. Among all ten endophytic bacterial isolates, four typical dominant morphotypes (Q2, Q5, Q6 and Q7) were chosen for molecular genetic identification.

#### 3.2.2. Pectolytic Activity Tests

Sterilized healthy potato tubers were cut into slices and placed in Petri dishes. Sterile 5 mm diameter paper discs were soaked in an overnight bacteria culture at 1 × 10^8^ CFU per ml for 10 min. The soaked disks were then placed on the potato slices. Petri dishes were incubated at a temperature of 25 °C for 2–3 days. Sterilized H_2_O was used as a control. The ability of strains to macerate potato tissue was observed over 7 days. Maceration of potato tissue or its absence gave information about the culture activity [96].

#### 3.2.3. Determination of Antifungal Activity

Determination of the antifungal activity of isolates against pathogens (*Fusarium* spp., *Alternaria* spp., *Botrytis cinerea* Pers., and *Sclerotinia sclerotiorum* (Lib.) de Bary) was performed using a modified perpendicular lines method (Figure 7).

Colonies of phytopathogenic fungi were placed on the PDA medium between sectorial perpendicular strokes of bacterial colonies. The degree of sensitivity of phytopathogenic fungi to endophytic bacteria was determined by the width of the zone of no growth [97,98].

Measurements were made according to indicators of linear growth of micromycete colonies. The width of the zone of no growth determined the degree of sensitivity of phytopathogenic fungi to endophytic bacteria. The radii of micromycete colonies were measured in four directions from their center. The ratio R_1_ to R_av_ was calculated, where R_1_–R_4_ are radii (mm) and R_av_ = (R_2_ + R_3_ + R_4_)/3.

### 3.3. Molecular Genetic Identification of Isolates

To perform sequencing analysis, bacteria were grown on potato glucose agar (PGA) for 48 h. Genomic DNA was isolated from the bacterial cell suspension using GeneJet Genomic DNA Purification Kit (ThermoScientific) according to the manufacturer’s protocol. The PCR reaction mix of 25 µL contained 12.5 µL of 2x DreamTaq PCR Master Mix (ThermoScientific), 30 pmol of each universal primer (27F: 5′-AGAGTTTGATCMTGGCTCAG-3′ and 1492r: 5′-CGGTTACCTTGTTACGACTT-3′), and 50 ng of DNA template. Amplification of 16S rDNA was carried out in Mastercycler Personal 5332 (Eppendorf) with the follow temperature cycles: the initial denature—95 °C, 2 min; 30 cycles—95 °C, 30 s; 55 °C, 45 s; 72 °C, 90 s; and the final elongation—72 °C, 7 min. The amplicon of ~1500 bp was run on a 1.7% agarose gel containing 0.01% of ethidium bromide and was visualized under ultraviolet light. Then the fragment was cut from the gel and purified with Zymoclean Gel DNA Recovery Kit (ZymoResearch, Irvine, CA, USA). DNA concentration was measured using DS -11 FX+ (DeNovix, Wilmington DE, USA). The sequencing reaction was carried out in both directions using a “BigDye Terminator v 3.1 Cycle Sequencing Kit” in Genetic Analyzer 3130 (Applied Biosystems, USA). The obtained nucleotide sequences were compared with those in GenBank with the help of NCBI Blastn (http://www.ncbi.nlm.nih.gov/blast), accessed on 20 November 2020. MEGA 11 [99] was used to perform the alignment of 16S rDNA sequences and phylogenetic analysis; dendrograms of genetic relationships were constructed using the Neighbor Joining method and Kimura’s 2-parameter model. The nucleotide sequences of 16S rRNA genes were downloaded from GenBank and used for the phylogenetic analysis. The accession numbers of the downloaded sequences are given in the parenthesis on the dendrograms (Figure 1, Figure 2 and Figure 3).

### 3.4. Biochemical Tests

#### 3.4.1. Methods of Sample Collection

One-year-old oak seedlings (n = 3) were used for complex phenolic phytochemical studies of leaves. Leaves after bacterial inoculation and control samples (without inoculation) were collected (n = 10) in June 2017 and 2018. The samples were dried in the oven (37 °C) to a constant weight and crushed. The resulting dry powder of the leaf mass was passed through a sieve number 40. To determine the total phenolic content, 1 g of the sample was added to 10 mL (1/10) of the methanol/water (80/20—*v*/*v*). To determine the total flavonoid content, the dry weight of leaves was extracted with 70% ethanol. The sample mixtures were extracted at 20 °C for 24 h. After that, the mixtures were centrifuged at 8000× *g* for 10 min, and the supernatant was taken for analysis. Prior to phytochemical analysis, the samples were stored in a freezer (–20 °C).

#### 3.4.2. Determination of the Total Phenolic Content in Leaves

The total content of phenolic compounds in the leaves was determined by spectrophotometry (SF Optizen Pop, Daejeon, Republic of Korea) using Folin-Ciocalteu’s phenol reagent [100]. An amount of 0.5 mL of Folin-Ciocalteu’s reagent was added to 0.1 mL of the extract. After 3 min we further added 0.4 mL of 1 M sodium carbonate solution (Na_2_CO_3_). The reaction mixture was kept in a dark place for 2 h in the thermostat at 23 °C, after which the spectrophotometry of samples was performed at λ = 760 nm in four analytical replicates. The calibration graph was based on a gallic acid.

#### 3.4.3. Determination of the Total Flavonoid Content

Quantitative content of flavonoids in leaf extracts was determined by spectrophotometry (using the Optizen Pop spectrophotometer, Daejeon, Republic of Korea) at λ = 415 nm. We added to 0.1 mL of leaf extract (1/10) a 0.3 mL of solution of aluminum chloride (AlCl_3_—25 g/L), 0.5 mL of sodium acetate (CH_3_COONa—100 g/L), and 4.1 mL bi-distilled water [100]. The blank sample contained 0.1 mL leaf extract and 4.9 mL bi-distilled water. The calibration graph was based on quercetin (Sigma, Hamburg, Germany).

#### 3.4.4. Determination of Antioxidant Activity

The concentration of phenolic antioxidants in the extracts was determined spectrophotometrically according to Brand-Williams using the stable free radical 2,2-diphenyl-1-picrylhydrazyl (DPPH•) [101,102]. Trolox ((±)-6-hydroxy2,5,7,8-tetramethyl-chroman-2-carboxylic acid, 97%) from Aldrich (Milwaukee, WI) was used as a standard to construct the calibration graph.

The reaction mixture contained 0.25 mL of plant extract, 1.75 mL of 80% ethanol, and 2 mL of 0.2 mM DPPH solution. In control samples, 2 mL of 80% ethanol was added to 2 mL of 0.2 mM DPPH solution. The test tubes were vigorously shaken and left for 30 min in the dark at room temperature. The optical density of the reaction mixture was determined at a wavelength of 515 nm. Inhibition (In) of DPPH a in percentage was calculated according to the formula:In DPPH = 100 (Dk − Do)/Dk,
in which:Dk—optical density in the absence of antioxidants (control);Do—optical density in the presence of antioxidants;

Antioxidant activity of plant extracts was expressed in μM-ekv Trolox.

The ratio of antioxidant activity values to the total content of phenols in leaves (AA*i*/Ph*i*) was determined using their relative values:AA*i* = AA*exp*/AA*c*;
Ph*i* = Ph*exp*/Phc,
where AA*i*—the ratio of the antioxidant activity of leaf extracts after bacterial inoculation (AA*exp*) to the control (AA*c*); Ph*i*—the ratio of the total content of phenols in the leaves after the inoculation of bacteria (Ph*exp*) to the control (Ph*c*).

#### 3.4.5. Biochemical Profiling of Plant Extracts with High-Performance Thin-Layer Chromatography

Biochemical profiling of oak leaf extracts was performed by the method of HPTLC on silica gel G60 plates (Merck Chemicals GmbH, Darmstadt, Germany). Separation of general phenolic compounds and flavonoids was performed in solvent systems: ethyl acetate—formic acid—acetic acid—water (*v*/*v*/*v*/*v*—100:11:11:25).

The standard (Quercetin, Rutin, Chlorogenic acid) solutions (3.0 µL of each concentration 1 mg·mL^−1^) were applied to the plates. The derivatization was performed with 0.5% NP reagent (1.0 g diphenylborinic acid aminoethyl ester dissolved in 200 mL ethyl acetate) and 1% PEG 400, followed by heating (5 min at 105 °C). Phenolic substances on the chromatogram were detected using UV light at 366 nm. The retention factor (Rf) of individual compounds was determined photodensitometrically using the software Sorbfil TLC ver. 2.3.0.2994 (JSC Sorbopolymer). The Rf value is equal to the distance traveled by the individual compound divided by the distance traveled by the mobile phase front (Appendix A, Figure A1 and Figure A2).

### 3.5. Field Testing of Bacteria on One-Year Seedlings of Q. robur

Field testing of bacteria was carried out on one-year seedlings of *Q. robur*. The plants were watered with the suspension of active isolates (Q2, Q7) and their mixture (Q2 + Q7) twice with a gap of three weeks. Additionally, after four weeks the leaves of annual oak plants were inoculated by endophytes and epiphytic and pathogenic bacteria in accordance with the scheme: a—control, b—*Bacillus subtilis* (Q2), c—*Bacillus amyloliquefaciens* (Q7), d—Q2/Q7 (mix 1:1), e—*Pectobacterium* spp., f—*Pseudomonas* spp., g—*Lelliottia amnigena* (Q5), h—*Delftia acidovorans* (Q6), i—epiphytic bacteria EpQ1, j—epiphytic bacteria EpQ2. Over the next months, observations of depigmentation of leaves and necrotic zones were observed.

### 3.6. Photo Documentation, Digital and Statistical Data Processing

Photo documentation and processing of digital images were performed with the specialized program Image-Pro Premier 9.0. The statistical significance of the differences between the values (*p* < 0.05) was determined by the analysis of the variance (ANOVA) method in the XLSTAT (Addinsoft Inc., New York, NY, USA, 2010). The data were compared using Tukey’s posterior test. The SigmaPlot 12.0 program was used for regression analysis.

## 4. Conclusions

Ten samples of endophytes were isolated from the tissues of immature acorn embryos of perennial trees *Quercus robur*, among which bacteria of the genus *Bacillus* were the dominant morphotypes. Based on the results of comparing the sequenced 16S rRNA genes (16S rDNA sequences) with the sequences deposited in GenBank, four types of endophytic bacteria were identified: *Bacillus amyloliquefaciens*, *Bacillus subtilis*, *Delftia acidovorans*, and *Lelliottia amnigena*. Endophytic dominants *Bacillus amyloliquefaciens* and *Bacillus subtilis* had different antagonistic activities in relation to phytopathogenic micromycetes that cause mycosis in oak plants. *Bacillus amyloliquefaciens* BSQ7-PSTQR-0920 showed high antifungal activity against phytopathogenic micromycetes *Fusarium tricinctum*, *Botrytis cinerea*, and *Sclerotinia sclerotiorum* using a modified method of microorganism antagonism analysis.

The isolation of endophytes from acorn tissues at the initial stages of their development and subsequent screening for pectinase activity made it possible to quickly obtain promising strains of potential PGP bacteria that are able to accelerate the regeneration of damaged tissues, have relatively high antifungal activity, and are tolerantly perceived by common oak seedlings.

*Bacillus amyloliquefaciens* and *Bacillus subtilis* bacteria inoculation was accompanied by a decrease in the total amount of polyphenols in the leaves of oak seedlings. Phytopathogenic bacteria *Pectobacterium* and *Pseuodomonas*, on the contrary, caused an increase in the concentration of polyphenols in plants by 2.0 and 2.2 times, respectively. An increase in the content of soluble phenols under the influence of biotic stress indicates the mobilization of the protective reserves of the plant organism; however, the index of the ratio of the antioxidants to phenols is an equally informative marker of the balance of the plant-microbial system. High values may indicate significant antioxidant potential of polyphenols in the leaves. The increased index of the ratio of antioxidant activity to the total content of phenols against the background of a decrease in the total pool of phenolic compounds in the leaves of oak seedlings, caused by inoculation with isolates of *Bacillus amyloliquefaciens* and *Bacillus subtilis*, may be a sign of the economic operation of the secondary synthesis system, which becomes possible under the conditions of a balanced plant-microbial system.

Therefore, the selection of natural endophytes of the PGPB class from the reproductive organs of certain plant species to create balanced plant-microbial systems within one genus or species has broad prospects, as it is based on the co-adaptation of a plant organism and a bacterial complex that has undergone long-term natural selection.

## Figures and Tables

**Figure 1 plants-12-01352-f001:**
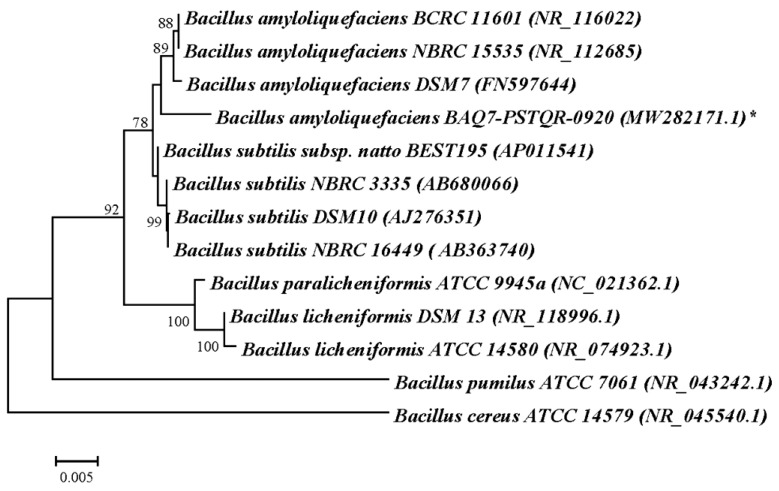
The dendrogram of the genetic similarity between strain BAQ7-PSTQR-0920 and various representatives of the *Bacillus* genus constructed on the basis of 16S rRNA gene sequences using the Neighbor-Joining method and Kimura’s 2-parameter model. The scale bar indicates 0.005 substitutions per nucleotide position. The analyzed strain is indicated by *.

**Figure 2 plants-12-01352-f002:**
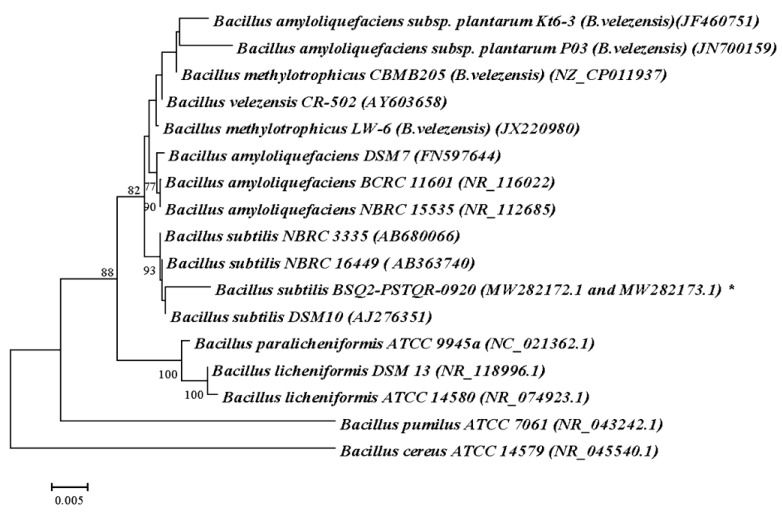
The phylogenetic relationships between different bacilli based on the 16S rDNA sequences. The dendrogram was built using the Neighbor-Joining method and Kimura’s 2-parameter model. The scale bar indicates 0.005 substitutions per nucleotide position. The analyzed strain is indicated by *.

**Figure 3 plants-12-01352-f003:**
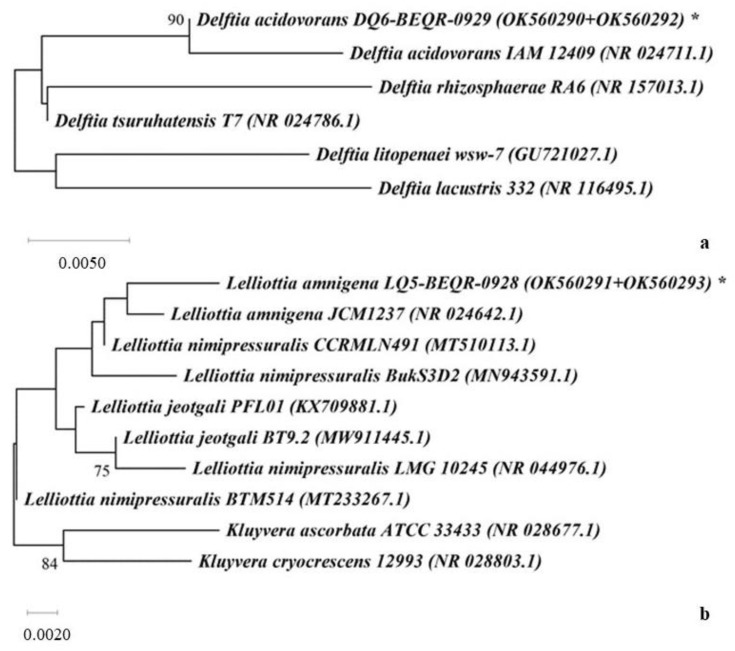
16S rRNA gene sequence-based dendrograms defining the phylogenetic position of the isolated bacterial strains DQ6-BEQR-0929 (**a**) and LQ5-BEQR-0928 (**b**) and their relationship with other close relative microorganisms. The isolated strains are indicated *.

**Figure 4 plants-12-01352-f004:**
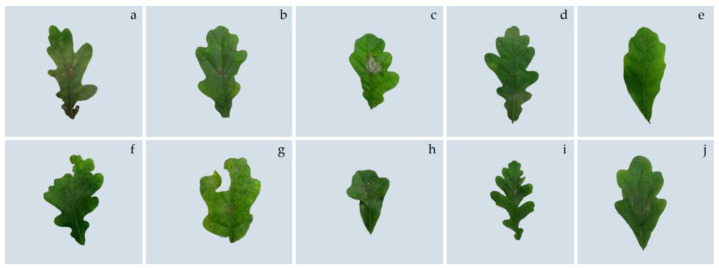
Inoculated leaves of annual oak plants by endophytes, epiphytic and pathogenic: (**a**)—control, (**b**)—*Pectobacterium* spp., (**c**)—*Pseudomonas* spp., (**d**)—*Bacillus subtilis* (Q2), (**e**)—*Bacillus amyloliquefaciens* (Q7), (**f**)—Q2/Q7 (mix 1:1), (**g**)—*Lelliottia amnigena* (Q5), (**h**)—*Delftia acidovorans* (Q6), (**i**)—EpQ1, (**j**)—EpQ2.

**Figure 5 plants-12-01352-f005:**
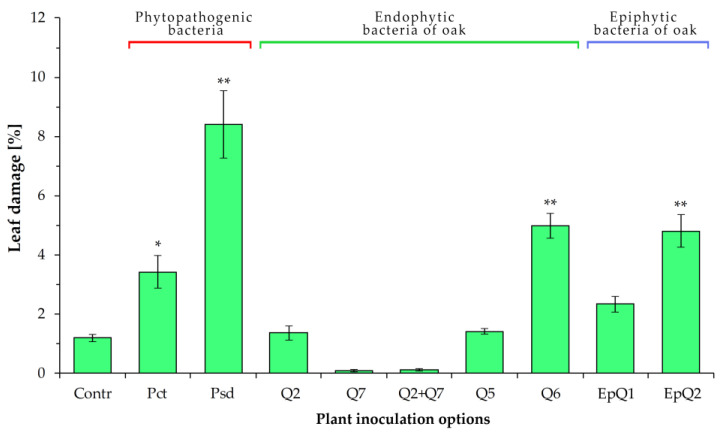
The degree of damage to the leaf surfaces of one-year-old oak seedlings one month after inoculation with bacterial isolates; the data were compared using Tukey’s test (HSD) by one-way ANOVA: *—significant differences to the Control at the level *p* < 0.05; **—at the level *p* < 0.01.

**Figure 6 plants-12-01352-f006:**
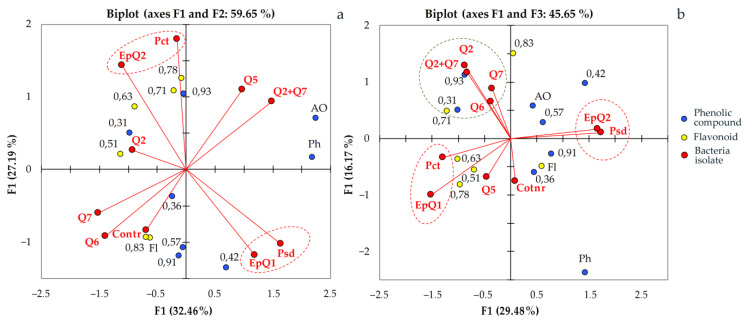
Principle component analysis by phytochemical profiles of phenolic compounds in common oak leaves after their treatment with endophytic bacteria.

**Figure 7 plants-12-01352-f007:**
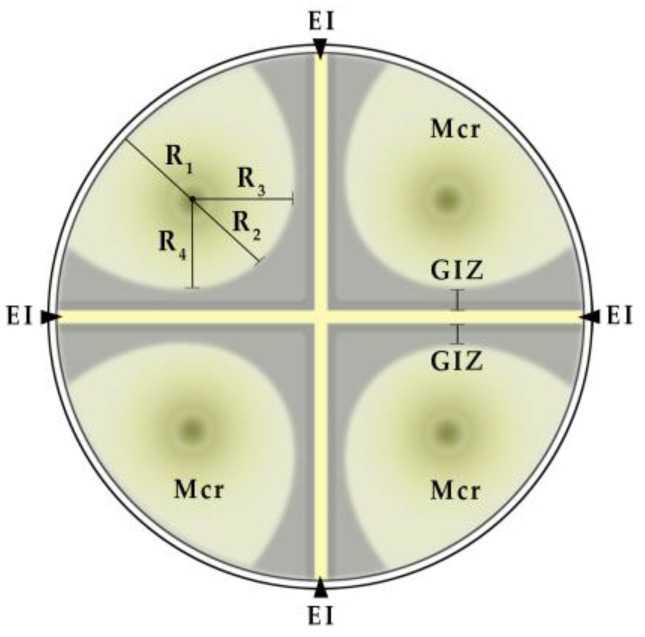
Scheme for evaluating the antifungal activity of endophytic bacteria based on indicators of linear growth of micromycete colonies: EI—endophytic isolate; GIZ—growth inhibition zone; Mcr—micromycete; R_1_–R_4_—micromycete colony radii.

**Table 2 plants-12-01352-t002:** Effect of endophytic isolate on linear growth parameters (mm) of micromycetes (x ± SE, n = 4).

Phytopathogens	EI	R_1_	R_2_	R_3_	R_4_	R_1_/R_av_	GIZ
*Botrytis*	Q2	20.3 ± 0.58	17.3 ± 1.00	16.3 ± 0.84 ^a^	17.7 ± 0.41	1.2 ± 0.08	1.3 ± 0.11
*cinerea*	Q7	22.2 ± 0.95	13.1 ± 0.19 ^b^	11.0 ± 0.25 ^b^	11.6 ± 0.32 ^b^	1.9 ± 0.10	8.0 ± 0.46
BCIBL-2143	Contr	8.8 ± 0.60	9.0 ± 0.65	8.87 ± 0.46	8.8 ± 0.20	1.0 ± 0.04	-
*Sclerotinia*	Q2	23.5 ± 1.10	15.8 ± 0.20 ^b^	13.4 ± 0.78 ^b^	13.2 ± 1.03 ^b^	1.7 ± 0.11	1.5 ± 0.21
*sclerotiorum*	Q7	22.7 ± 1.79	16.6 ± 0.37 ^b^	14.7 ± 0.47 ^b^	13.8 ± 0.24 ^b^	1.5 ± 0.12	4.0 ± 0.26
SIBL-2135	Contr	20.5 ± 1.63	21.3 ± 0.86	18.4 ± 1.93	18.9 ± 1.22	1.1 ± 0.05	-
*Fusarium*	Q2	19.5 ± 0.47	13.6 ± 0.53 ^a^	12.9 ± 0.61 ^a^	12.7 ± 0.56 ^b^	1.5 ± 0.08	5.1 ± 0.48
*tricinctum*	Q7	18.2 ± 1.23	7.2 ± 0.39 ^b^	7.4 ± 0.24 ^b^	7.4 ± 0.40 ^b^	2.5 ± 0.16	13.2 ± 0.36
FTIBL-2151	Contr	20.6 ± 0.85	20.0 ± 0.49	19.7 ± 0.20	19.4 ± 0.71	1.0 ± 0.03	-

Note: R_1_–R_4_—radii (mm); R_av_ = (R_2_ + R_3_ + R_4_)/3; GIZ—growth inhibition zone (mm); the data were compared using Tukey’s test (HSD) by one-way ANOVA: ^a^—significant differences to the R_1_ at the level *p* < 0.05; ^b^—at the level *p* < 0.01.

**Table 3 plants-12-01352-t003:** The content of phenolic compounds in the leaves of common oak plants after inoculation with endophytic bacteria (x ± SE, n = 3).

Samples	Phenols, mg·g^−1^		Flavonoids, mg·g^−1^		Antioxidants, μM-eq		AA*i*/Ph*i* *
	Samples	Experim	Samples	Experim	Samples	Experim	Experim
Control	101.3 ± 1.51	125.7 ± 3.84	3.8 ± 0.14	2.0 ± 0.08	196.0 ± 7.39	187.1 ± 5.72	1.00 ± 0.01
Q2	98.9 ± 3.02	102.0 ± 3.40	2.7 ± 0.10 ^b^	1.8 ± 0.06 ^b^	210.5 ± 7.94	242.1 ± 7.40	1.59 ± 0.01 ^b^
Q7	84.3 ± 2.78	77.8 ± 2.59	1.9 ± 0.07 ^b^	1.7 ± 0.06 ^b^	174.5 ± 6.58	164.8 ± 5.49	1.44 ± 0.01 ^b^
Q2/Q7 (mix 1:1)	143.6 ± 4.74 ^b^	104.4 ± 3.48	2.3 ± 0.07 ^b^	1.9 ± 0.06 ^b^	252.2 ± 9.51 ^a^	232.6 ± 7.75	1.51 ± 0.01 ^b^
*Pectobacterium*	140.5 ± 4.64 ^b^	157.3 ± 5.24 ^b^	2.4 ± 0.09 ^b^	2.3 ± 0.08 ^b^	259.6 ± 7.94 ^b^	254.1 ± 8.47 ^b^	1.10 ± 0.01 ^a^
*Pseuodomonas*	150.2 ± 4.96 ^b^	175.9 ± 5.86 ^b^	2.3 ± 0.07 ^b^	1.7 ± 0.06 ^b^	257.7 ± 9.72 ^b^	256.5 ± 8.55 ^b^	0.99 ± 0.01
Q5	173.0 ± 5.71 ^b^	194.3 ± 6.48 ^b^	0.4 ± 0.02 ^b^	0.3 ± 0.01	257.5 ± 7.87 ^b^	246.3 ± 8.21 ^a^	0.86 ± 0.01 ^b^
Q6	118.8 ± 3.92	92.7 ± 3.09	3.2 ± 0.12 ^a^	2.2 ± 0.07 ^b^	201.4 ± 7.60	163.9 ± 5.46	1.20 ± 0.01 ^b^
EpQ1	143.0 ± 4.72 ^b^	158.3 ± 5.28 ^b^	3.4 ± 0.13 ^b^	3.4 ± 0.11 ^b^	242.6 ± 9.15	238.6 ± 7.95	1.02 ± 0.01
EpQ2	165.9 ± 5.47 ^b^	173.6 ± 5.79 ^b^	2.5 ± 0.08 ^b^	2.4 ± 0.08 ^b^	245.7 ± 7.51	247.5 ± 9.33 ^a^	0.96 ± 0.01

*—the ratio in the experiment of the amount of antioxidants (AA*i*) to the amount of total phenols (Ph*i*); the data were compared using Tukey’s test (HSD) by one-way ANOVA: ^a^—significant differences to the Control at the level *p* < 0.05; ^b^—at the level *p* < 0.01.

## Data Availability

Not Applicable.

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
