# Peer review of "Antifungal Activity and Effect of Plant-Associated Bacteria on Phenolic Synthesis of Quercus robur L."

_plants, 2023, doi:10.3390/plants12061352_

Round 1

Reviewer 1 Report

Paper reports an interesting research on the antifungal activity of endophytic bacteria associated with oak trees. Results have a great utility value. It should be emphasized, that the paper is quite well written, which shows good organization. However, a lot of sentences should be reedited and improved. English should be thoroughly improved thru ought the manuscript. In my opinion, the manuscript deserve to be published in Plants, an MDPI journal.

My general comments are as follows:

1.     Line 70 – the sentence seems to be unfinished.

2.     Line 110 – Authors should use the term microbiota instead of microflora.

3.     Lines 114-116 – sentence should be reedited to show clearly the purpose of the conducted research. English should be improved.

4.     Lines 124-126, 131-132, 150 etc. – sentences should be reedited. English should be improved.

5.     Lines 332 – 333, 340 etc. – Latin names of microorganisms should be written in italics

6.     Latin names of microorganisms should be written in full when mentioned first time; next should be abbreviated, e.g. Chapter 4.

7.     Section conclusions is too long. It should be shortened

Author Response

Response to Reviewer 1 Comments 

Dear Reviewer,

First of all, we sincerely thank you for your valuable suggestions on this article, from which the authors have learned a lot of new knowledge and have carefully considered and revised it. After adjustment by the author, some mistakes in the materials and methods have been placed, and the errors in the chapter names and tables have been changed. Therefore, the number of lines in the article may not match the previous one, after proofreading one by one, the author carefully compared the number of lines before and after. The response to the review of this article is as follows.

Point 1: Line 70 – the sentence seems to be unfinished.

Response 1: Thanks for the expert advice, the author missed one word in the end of the sentence ”studied”.

Point 2: Line 110 – Authors should use the term microbiota instead of microflora.

Response 2: Thanks for the advice, the authors corrected it.

Point 3: Lines 114-116 – the sentence should be re-edited to show clearly the purpose of the conducted research. English should be improved.

Response 3: Here the aim was to isolate and identify endophytic bacteria from the tissues of im-mature Quercus robur L. acorns and to study their biological activity about pathogenic micromycetes and seedlings of Q. robur.

Point 4: Lines 124-126, 131-132, 150 etc. – sentences should be reedited. English should be improved.

Response 4: Thanks for the expert advice, the authors rethought the sentences and changed it to "The objects of research were isolates of endophytic microorganisms isolated from the tissues of immature oak acorns and cultures of phytopathogenic micromycetes pathogens of woody plants."

“The isolation of endophytic bacteria from the tissues of embryos of immature oak acorns [79] and epiphytic bacteria from leaves was performed by used methods as reported in Borkar [80] and phytopathogenic bacteria research methods [81].”

“2.3. Determination of the degree of sensitivity of phytopathogenic isolates”

Point 5: Lines 332 – 333, 340 etc. – Latin names of microorganisms should be written in italics

Response 5: Thank you for the expert's reminder. This is the author's writing error, now the author has written all the Latin names of microorganisms in the text in italics.

Point 6: Latin names of microorganisms should be written in full when mentioned the first time; next should be abbreviated, e.g. Chapter 4

Response 6: Thank you for your comments, the question regarding the abbreviation will be reviewed and corrected.

Point 7: The section conclusions is too long. It should be shortened

Response 7: The text is abbreviated.

Reviewer 2 Report

The authors in their manuscript entitled “Antifungal Activity and Effect of Plant-Associated Bacteria on 2 Phenolic Synthesis of Quercus robur L.”, study the effect endophytic bacteria from acorns of Quercus robur L. against pathogenic fungi and on seedlings of Q. robur.

The argument is clearly set, introduction sufficiently gives the background, the methods applied are correct and results -to a certain extent- are well presented and adequately discussed, however, there are points that probably need further clarification, amendment, or addition. Additional experiments seem not to be required, provided that the manuscript reforms and explanations (as described below) once presented by the authors in the revised manuscript succeed to support the arguments.

In specific:

11.       English language expression, syntax, spelling and typographic, major, or minor (e.g., italics in scientific names in most of the cases) issues should be corrected throughout the whole text. Please check Lines: 34, 65-70 (syntax and expression), 98, 124 (other word instead of “objects”), 134 (“by”), 150, 179 (“gene(s)”). Check Lines: 290, 291, 300, 302-304, 310, 312-319, 332,333, 340, 341, 354, 356, 357, 367, 371-373, 393, 431, 432, 439 and furthermore throughout the manuscript  for Italics.

22.       In Table 1 please correct the alignment of the third column in order to have the data presented correctly

33.      In Materials and Methods

a.       Define “Centuries-old” in line119. Though the meaning is understood, there should be a more precise description (i.e., 2-centuries or 10-centuries old)

b.       The 2.2.2 Paragraph seems incomplete. There is a description of the process to a certain extend but there is no description of what is measured.

c.       Paragraphs 2.3.3 and 2.3 fall under the same concept. Please join and reform numbering to 2.3

d.       In Figure 1 please enlarge the letters for radii etc. on the Petri dish. On the figure but also further down in the manuscript the authors use the concept “PGPB, Plant Growth Promoting Bacteria” to describe their isolates, however, they have not shown experimentally that their isolates act as growth promoters of plants. Thus, this is actually an assumption made (originating probably from similar bibliography reports) rather than a fact experimentally validated. Please choose another word/concept (i.e., “endophytic isolates?”) to refer to the bacteria tested in this work. Please then also define where the bacteria area on the Petri dish is located. It is not clear if they are on the crossed lines or spread throughout the whole of the P. dish. Revise the figure legend accordingly for PGPBs.

e.       In line 180, do the authors refer to the sequences downloaded from NCBI? If so, please site here all the sequences used which also appear in figures 2, 3 and 4

f.        Paragraphs 2.4.2 to 2.4.6 do not fall under the general term “2.4 Molecular genetic identification of isolates”. They are actually biochemical assays thus please number them individually (i.e., 2.5, 2.6, etc.)

44.       In the results section

a.       In lines 252-253 the authors refer to “Ten samples”. Please use the term “isolates” instead of “samples” since this may confuse. There should also be defined from how many isolates (the total number of which) tested, these 10 selected isolates have been chosen. This could be also declared in the M and M section as well.

b.       Are all these 10 isolates sequenced? Please define, and if so, the relevant sequences should be submitted to GenBank and reported in paragraph 3.2 as well. A relevant table or text, presenting the codified names (Q1, Q2, etc. ) with the full names for simplification, would be of help for quick reference.

c.       In lines 313-314 there is a report for IAA production. Do the authors refer to bibliographic data or to their own isolates under test. If the former then, the reference that the isolates produce IAA is an assumption (if not shown experimentally) and the relevant parts from Summary, Results and Conclusions should be withdraw. If the latter then the relevant experimental part and results should be presented and discussed.

d.       From line 330 onwards the text should be devided in specific paragraphs with the relevant titles, since they describe different experiments and do not fall under paragraph 3.2 as a whole. Please devide the text accordingly (i.e., lines 330-362, 363-391, 392-412, and so on in order to give a clear image of the experiments discussed.

e.       Table 2 should be reformed. Please check the term PGPB also here. Please Define what BCIBL, SIBL and FTIBL are.

f.        The paragraph between lines 364-370 seems not connected to the rest of the text. If there is a reason for referring to this work(s) please specify and try to connect it to your results in the discussion.

g.       The experiment described in lines 371 onwards is not described in the M & M section. Please define.

h.       The concept in lines 382-386 seems rather an assumption since no experimental data are presented to validate the biological action of these compounds identified.

i.         In the same experiment it is not clear how the experimental set up was conducted and which was the actual aim. If the aim is to compare with the effect of Pct and Psd, this should be defined. There should be also a clarification of cause of leaf damage in the Control sample comes from. Is it mechanical? Is it a healthy leaf? If the latter the reason for the damage should be explained.

j.          A suggestion would be to present in order: the Control, Pct, Psd, and then all the rest in both figures 5 and 6, to facilitate more obvious comparisons

k.       In table 3 the authors should define what the “Sumples” is and why are they used-presented. Are they different from “Experim” and if so to what?

55.      A relevant adaptation of the conclusions section should be made according to reforms in the rest of the Results and Discussion section.

Author Response

Response to Reviewer 2 Comments

Dear Reviewer,

First of all, we sincerely thank you for your valuable suggestions on this article, from which the authors have learned much new knowledge and carefully considered and revised it. After adjustment by the author, some mistakes in the materials and methods have been placed, and the errors in the chapter names and tables have been changed. Therefore, the number of lines in the article may not match the previous one. After proofreading one by one, the author carefully compared the number of lines before and after. The response to the review of this article is as follows.

Point 1: English language expression, syntax, spelling and typographic, major, or minor (e.g., italics in scientific names in most of cases) issues should be corrected throughout the whole text. Please check Lines: 34, 65-70 (syntax and expression), 98, 124 (other words instead of “objects”), 134 (“by”), 150, 179 (“gene(s)”). Check Lines: 290, 291, 300, 302-304, 310, 312-319, 332,333, 340, 341, 354, 356, 357, 367, 371-373, 393, 431, 432, 439 and furthermore throughout the manuscript for Italics.

Response 1: Thanks for the expert advice. The author corrected sentences, some syntax mistakes and expressions in particular: Lines: 34, 65-70 (italics), 98, 124 (In research were used), 134 (according), 150 (Determination of degree sensitivity of phytopathogenic isolates), 179 (genes).

In addition, we checked spelling and typographic italics in scientific names in particular Lines: 290, 291, 300, 302-304, 310, 312-319, 332,333, 340, 341, 354, 356, 357, 367, 371-373, 393, 431, 432, 439 and looked throughout the manuscript for Italics

Point 2:  In Table 1, please correct the alignment of the third column in order to have the data presented correctly

Response 2: Thanks for the correction. The authors corrected In Table 1 columns in order.

Point 3: In Materials and Methods

  1. Define “Centuries-old robur trees ” in line119. Though the meaning is understood, there should be a more precise description (i.e., 2-centuries or 10-centuries old).
  2. The 2.2.2 Paragraph seems incomplete. There is a description of the process to a certain extend but there is no description of what is measured.
  3. Paragraphs 2.3.3 and 2.3 fall under the same concept. Please join and reform numbering to 2.3.
  4. In Figure 1 please enlarge the letters for radii etc. on the Petri dish. On the figure but also further down in the manuscript the authors use the concept “PGPB, Plant Growth Promoting Bacteria” to describe their isolates, however, they have not shown experimentally that their isolates act as growth promoters of plants. Thus, this is actually an assumption made (originating probably from similar bibliography reports) rather than a fact experimentally validated. Please choose another word/concept (i.e., “endophytic isolates?”) to refer to the bacteria tested in this work. Please then also define where the bacteria area on the Petri dish is located. It is not clear if they are on the crossed lines or spread throughout the whole of the P. dish. Revise the figure legend accordingly for PGPBs.
  5. In line 180, do the authors refer to the sequences downloaded from NCBI? If so, please site here all the sequences used which also appear in figures 2, 3 and 4.
  6. Paragraphs 2.4.2 to 2.4.6 do not fall under the general term “2.4 Molecular genetic identification of isolates”. They are actually biochemical assays thus please number them individually (i.e., 2.5, 2.6, etc.)

Response 3: in Materials and Methods

  1. Clarified about certain age of Q. robur trees (age 200 years old).
  2. The paragraph has been completed “The ability of strains to macerate of potato tissue during 7 days was observed. Maceration of potato tissue or its absence indicated the culture activity”.
  3. The paragraph has been corrected in particular: Paragraphs 2.2.3. Determination of antifungal activity combine with 2.2. Isolation, cultivation and studying of the biological activity of endophytic bacteria; than 2.4. Biochemical tests; 2.5. Photo documentation, digital and statistical data processing.
  4. In Figure 1, put some corrections particularly, EI – endophytic isolate.
  5. The authors changed the explanation and indicated it in such a way “The nucleotide sequences of 16S rRNA gene were downloaded from GenBank and used for the phylogenetic analysis. The accession numbers of the downloaded sequences were given in the parenthesis on the dendrograms (figures 2, 3, 4)”.
  6. Thank you for the correction. The Paragraphs about biochemical research were moved and stated in the correct order.

Point 4: 44. In the results section

  1. In lines 252-253 the authors refer to “Ten samples”. Please use the term “isolates” instead of “samples” since this may confuse. There should also be defined from how many isolates (the total number of which) tested, these 10 selected isolates have been chosen. This could be also declared in the M and M section as well.
  2. Are all these 10 isolates sequenced? Please define, and if so, the relevant sequences should be submitted to GenBank and reported in paragraph 3.2 as well. A relevant table or text, presenting the codified names (Q1, Q2, etc.) with the full names for simplification, would be of help for quick reference.
  3. In lines 313-314 there is a report for IAA production. Do the authors refer to bibliographic data or to their own isolates under test. If the former then, the reference that the isolates produce IAA is an assumption (if not shown experimentally) and the relevant parts from Summary, Results and Conclusions should be withdraw. If the latter then the relevant experimental part and results should be presented and discussed.
  4. From line 330 onwards the text should be devided in specific paragraphs with the relevant titles, since they describe different experiments and do not fall under paragraph 3.2 as a whole. Please devide the text accordingly (i.e., lines 330-362, 363-391, 392-412, and so on in order to give a clear image of the experiments discussed.
  5. Table 2 should be reformed. Please check the term PGPB also here. Please Define what BCIBL, SIBL and FTIBL are.
  6. The paragraph between lines 364-370 seems not connected to the rest of the text. If there is a reason for referring to this work(s) please specify and try to connect it to your results in the discussion.
  7. During field testing of isolated bacteria on one-year seedlings of robur, it was established that after inoculation with phytopathogenic bacteria Pectobacterium spp. and Pseudomonas spp. over time, quite significant lesions with characteristic depigmentation and necrotic zones were formed on the leaves (Fig. 5).
  8. The concept in lines 382-386 seems rather an assumption since no experimental data are presented to validate the biological action of these compounds identified.
  9. In the same experiment it is not clear how the experimental set up was conducted and which was the actual aim. If the aim is to compare with the effect of Pct and Psd, this should be defined. There should be also a clarification of cause of leaf damage in the Control sample comes from. Is it mechanical? Is it a healthy leaf? If the latter the reason for the damage should be explained.
  10. A suggestion would be to present in order: the Control, Pct, Psd, and then all the rest in both figures 5 and 6, to facilitate more obvious comparisons.
  11. In table 3 the authors should define what the “Sumples” is and why are they used-presented. Are they different from “Experim” and if so to what?

Response 4:

  1. Thanks for the expert advice, the authors rethought the sentences and changed them.

The isolation of endophytic bacteria from the tissues of embryos of immature oak acorns [79], and epiphytic bacteria (EpQ1, EpQ2) from leaves was performed as reported in Borkar [80] and phytopathogenic bacteria research methods [81]. Bacteria were cultivated on the PDA medium (Potato Dextrose Agar).The bacteria were Gram-stained and used to study of the morphological features of bacteria. Also, spore formations, the character of growth on solid and liquid media of the endophytic microorganisms were studied. Isolates of endophytic microorganisms were identified by morphological and cultural properties according to generally recognized methods in bacteriology [81]. Among all of ten endophytic bacterial isolates four typical dominant morphotypes (Q2, Q5, Q6 and Q7) were chosen for molecular genetic identification.

  1. Thank you for the expert's opinion. Particularly in punkt 3.1 it indicates that «For the identification and study of ecophysiological properties, 4 typical representatives of the dominant morphotypes were chosen: bacterial isolates Q2, Q5, Q6 and Q7.» (lines 260 – 266).
  2. Yes, this is indeed an assumption based on literature data. The corresponding link has been added. (Shao J. et al. 2020; Keswani et al. 2020)
  3. Relevant subdivisions have been added.
  4. Thank you for a valid recommendation. The authors deleted the paragraph between lines 364-370.
  5. Thank you for the expert's opinion. The authors corrected some calculations.

During growth in the control variant of Sclerotinia sclerotiorum, an undulating growth of light thin mycelium hyphae, more compacted towards the centre, was observed. The authors also added supplementary material “Appendix A, Figure S1”, where can observe this.

Sclerotinia

sclerotiorum

SIBL-2135

Q2

23,5 ± 1,10

15,8 ± 0,20 b

 13,4 ± 0,78 b

13,2 ± 1,03 b

1,7 ± 0,11

1,5 ± 0,21

Q7

22,7 ± 1,79

16,6 ± 0,37 b

 14,7 ± 0,47 b

13,8 ± 0,24 b

1,5 ± 0,12

4,0 ± 0,26

Contr

20,5 ± 1,63

21,3 ± 0,86

18,4 ± 1,93

18,9 ± 1,22

1,1 ± 0,05

-

  1. I'm very sorry, this is the author's mistake. We added information about field testing of isolated bacteria on one-year seedlings of Q. robur in 2 punkt materials and methods. “Field testing of bacteria was carried out on one-year seedlings of Q. robur. The plants were watered of the suspension of active isolates (Q2, Q7) and their mixture (Q2+Q7) twice with a gap of three weeks. Also after four weeks the leaves of annual oak plants were inoculated by endophytes, epiphytic and pathogenic bacteria in accordance with the scheme: a - control, b – Bacillus subtilis (Q2), c – Bacillus amyloliquefaciens (Q7), d – Q2/Q7 (mix 1:1), e – Pectobacterium, f – Pseudomonas spp., g – Lelliottia amnigena (Q5), h – Delftiatia acidovorans (Q6), i – epiphytic bacteria EpQ1, j – epiphytic bacteria EpQ2. Over the next months, observations of depigmentation of leaves and necrotic zones were observed”. In the third punkt all information are available.
  2. Questions about assumptions or specific facts. Indeed, in this context, explaining the effect is only a guess. We have made appropriate changes in the text “One possible explanation of this effect is obvious that the impact of rapid and complete regeneration of leaf tissues is possible due to the action of bioactive compounds, in particular, phytohormones, which are synthesized by endophytic bacteria. However, further studies are needed to confirm this assumption.”
  3. Thank you for your response. This will really improve the perception of the presented material. Accordingly, an additional explanation is included in the text.
  4. It was changed on: «b – Pectobacterium spp., c – Pseudomonas spp., d – Bacillus subtilis (Q2), e – Bacillus amyloliquefaciens (Q7), f – Q2/Q7 (mix 1:1), g – Lelliottia amnigena (Q5), h – Delftiatia acidovorans (Q6), i – epiphytic bacteria EpQ1, j – epiphytic bacteria EpQ2.»
  5. Thanks for the expert reminder. This is the author's writing error. The word was correct “Samples”.

Point 5: Line 55. A relevant adaptation of the conclusions section should be made according to reforms in the rest of the Results and Discussion section.

Response 5: Thank you for the expert's reminder. 

Reviewer 3 Report

According to the idea, this is a very new and relevant work, but the design and presentation leave much to be desired.

The abstract mentions PGPTs(?), the ability of the studied bacteria to synthesize phytohormones and affect Alternaria, among other things. These results are not presented in the paper.

The rules give the sequence of presentation of materials: introduction, results, discussion, materials and methods.

The design of the experiment also raises questions.

How do endophytic bacteria from acorns and endophytes of oak plants correlate? Is there a vertical transmission of them? How do the endophytes of the root, stem, and branches of oak plants differ?

Fragments of the sequenced 16S rRNA gene for the BSQ2 strain at 564 and 511 bp, for LQ5 – 1061 bp, PQ6 -801 bp. why are they so short? Is their identification reliable in this case?

29: cinerea in italics

34-35: You are talking about the use of bacteria as biopesticides. Give advantages and disadvantages in comparison with chemical pesticides and agrotechnical treatments. How they can supplement or replace them.

98: Arthrobacter in italics?

139: degree sign in upper case

3.2. It is customary to write the names of organisms in Latin in italics. For example, L. Amnigena, L. jeotgal, Paenibacillus.

290, 291, 295 Names in Latin in italics.

307-308 About the summaty of what properties are we talking about (a specific result)?

314 PGPTs-? or PGPBs?

332: Italics: Fusarium spp., Alternaria spp., Rhizoctonia spp., Verticillium spp., Botrytis spp., Sclerotinia spp.

371-374 Which phytopathogenic microorganisms were used? There is no mention of them in the materials and methods.

Fig. 5 and Fig. 6 EpQ1 and EpQ2 what are the strains? The history of their origin is not clear.

Table 2. Mushroom growth in the control is very weak. The R values in the control are lower than with bacteria. Why?

Table 3. Sumples?

mg g-1

The text mentions both phytohormonal and nitrogenase activity. Studied strains have these abilities?

I would like to see the characteristic TLC plates in the form of a drawing. What are these flavonoids with Rf 0.83, 0.71, 0.78, 0.42, 0.17, etc. Is there this data?

453:PGPM, try to use one thing. Too free usage of abriveatures.

558: PGP try to use one thing. Too free usage of abriveatures.

523 ISR – abbreviation

The work was carried out (leaf samples were taken) in 2017-18, and funding was in 2021-22?

As a wish for the future. It would be interesting to see how the studied bacteria affect the hormonal balance of plants. The article describes the effect on the shape of leaves and the concentration of phenolic compounds as stress markers. At the same time, their concentrations can change in any direction, regardless of whether the plant is under stress or not. A comprehensive assessment of the content of phenolic compounds, the hormonal background of plants, and growth characteristics is necessary.

Author Response

Response to Reviewer 3 Comments

Dear Reviewer,

First of all, we sincerely thank you for your valuable suggestions on this article, from which the authors have learned a lot of new knowledge and have carefully considered and revised it.

After adjustment by the author, some mistakes in the materials and methods have been placed, and the errors in the chapter names and tables have been changed. Therefore, the number of lines in the article may not match the previous one, after proofreading one by one, the author carefully compared the number of lines before and after. The response to the review of this article is as follows.

Point 1:

How do endophytic bacteria from acorns and endophytes of oak plants correlate? Is there a vertical transmission of them? How do the endophytes of the root, stem, and branches of oak plants differ?

Response 1: Thank you for your question and interest in clarification.

Indeed, now the endophytic bacteria topic is very relevant. Knowledge of the mechanisms of microbial inheritance in plants and a greater understanding of their functional role in plant fitness can lead to the development of strategies for breeding crops with greater disease resistance and adapted to stressful environments. In particular, Abdelfattah, A. et al. 2021 give a whole list of advantages (Abdelfattah, A., Wisniewski, M., Schena, L. and Tack, A.J.M. (2021), Experimental evidence of microbial inheritance in plants and transmission routes from seed to phyllosphere and root. Environ Microbiol, 23: 2199-2214. https://doi.org/10.1111/1462-2920.15392). In this article they tested the hypothesis that the plant microbiome is partially inherited through vertical transmission. And also that the microbial communities associated with the acorn's embryo and pericarp and the developing seeding's phyllosphere and root systemsÑŽ

The collective data provide clear evidence of vertical transmission from seed to seedling and highlight the important role of vertical transmission during the assembly of the plant microbiome.

Point 2:  Fragments of the sequenced 16S rRNA gene for the BSQ2 strain at 564 and 511 bp, for LQ5 – 1061 bp, PQ6 -801 bp. why are they so short? Is their identification reliable in this case?

Response 2: Thank you for your question and interest in this question. The species identification was performed using set of characteristics. It was carried out based on some morphological characteristics (Gram, cytology) and 16S rDNA sequencing including phylogenetic analysis. The variable part of 16S rRNA gene was sequenced in all isolates, so it gave a relevant information for their identification. All sequences were compared with the 16S rDNA sequences of type strains and the variable nucleotides were compared between all samples belonging to the same genus.

Besides, BAQ7 and BSQ2, 1454 and 1075 bp (in total), respectively, belong to Bacillus subtilis group and to identify these bacteria even 500-600 nucleotide from 5’-end can be enough according to [Reva, O. N., Dixelius, C., Meijer, J., & Priest, F. G. (2004). Taxonomic characterization and plant colonizing abilities of some bacteria related to Bacillus amyloliquefaciens and Bacillus subtilis. FEMS microbiology ecology, 48(2), 249-259.].

Point 3: Line 29: cinerea in italics

Response 3: I am very sorry. Due to an error when the article was uploaded for the first time some Latin names didn't transfer in italic.This is corrected.

Point 4: Line 34-35: You are talking about the use of bacteria as biopesticides. Give advantages and disadvantages in comparison with chemical pesticides and agrotechnical treatments. How they can supplement or replace them.

Response 4: Thank you for your question and interest in this problem. Endophytic bacteria and fungi provide plethora of bioactive molecules, which can act as an inhibiting agents including QS quenching enzymes such as lactonase, acyclase, and QS inhibitor molecules (LaSarre and Federle, 2013). Such agents can provide promising approach to control phytopathogens and suppress virulence expression in them. They assist in degrading quorum-sensing signals from pathogenic microbes and disrupt intercellular communication (Rutherford and Bassler, 2012). In particular Paz et al 2018, shoves as alternative, novel methods to manage plant diseases, endophytic microorganisms. The advantage of endophytic bacteria, in the case of their successful integration into the plant-microbial system, is a mild, long-lasting and continuous action.

Point 5: Line 98: Arthrobacter in italics?

Response 5: I am very sorry. Due to an error when the article was uploaded for the first time some Latin names didn't transfer in italic.This is corrected.

Point 6: Line 139: degree sign in upper case

Response 6: I am very sorry for this mistake. This is corrected in the text.

Point 7: 3.2. It is customary to write the names of organisms in Latin in italics. For example, L. Amnigena, L. jeotgal, Paenibacillus.

Response 7: I am very sorry. Due to an error when the article was uploaded for the first time some Latin names didn't transfer in italic.This is corrected.

Point 8: Line 314 PGPTs-? or PGPBs?

Response 8: I am very sorry. This is corrected on PGPBs.

Point 9: Line 332: Italics: Fusarium spp., Alternaria spp., Rhizoctonia spp., Verticillium spp., Botrytis spp., Sclerotinia spp.

Response 9: I am very sorry. Due to an error when the article was uploaded for the first time some Latin names didn't transfer in italic.This is corrected.

Point 10: Line 371-374 Which phytopathogenic microorganisms were used? There is no mention of them in the materials and methods.

Response 10: Thanks for the expert reminder. The authors show some information about this in punkt 2.2.1. Isolation and cultivation of bacteria

The isolation of endophytic bacteria from the tissues of embryos of immature oak acorns [79], and epiphytic bacteria (EpQ1,  EpQ2) from leaves was performed as reported in Borkar [80] and phytopathogenic bacteria (Pectobacterium spp., Pseudomonas spp.) research methods [81].

But the authors also added and describe this information in the second punkt matherials and methods, “2.5. Field testing of bacteria on one-year seedlings of Q. robur”.

“Field testing of bacteria was carried out on one-year seedlings of Q. robur.

The plants were watered of the suspension of active isolates (Q2, Q7) and their mixture (Q2+Q7) twice with a gap of three weeks. Also after four weeks the leaves of annual oak plants were inoculated by endophytes, epiphytic and pathogenic bacteria in accordance with the scheme: a - control, b – Bacillus subtilis (Q2), c – Bacillus amyloliquefaciens (Q7), d – Q2/Q7 (mix 1:1), e – Pectobacterium spp., f – Pseudomonas spp., g – Lelliottia amnigena (Q5), h – Delftiatia acidovorans (Q6), i – epiphytic bacteria EpQ1, j –  epiphytic bacteria EpQ2. Over the next months, observations of depigmentation of leaves and necrotic zones were observed”.

Point 11: Fig. 5 and Fig. 6 EpQ1 and EpQ2 what are the strains? The history of their origin is not clear.

Response 11: Thanks for the expert reminder. The authors ordered numeration in Fig.5 and improved Fig. 6.

Point 12: Table 2. Mushroom growth in the control is very weak. The R values in the control are lower than with bacteria. Why?

Response 12: Thanks for the expert reminder. The growth of the mycelium of Botrytis cinerea on the surface of the nutrient medium in the control occurred slower, through abundant sporulation and spread of the micromycete over the entire surface of the medium. At the same time, during growth in the presence of endophytic bacteria, especially the more active Q7 isolate, a delay in the intensity of sporulation and limitation of mycelium growth were noted.

During growth in the control variant of Sclerotinia sclerotiorum, an undulating growth of light thin mycelium hyphae, more compacted towards the center, was observed. The authors also added supplementary material “AppendixA , Figure S1”, where can observe this.

Point 13: Table 3. Sumples

Response 13: I am very sorry for this mistake. We are correct on “Samples”.

Point 14: Table 3 “mg·g-1

Response 14: I am very sorry for this mistake. We are correct on “mg·g-1”.

Point 15: The text mentions both phytohormonal and nitrogenase activity. Studied strains have these abilities?

Response 15: Thank you for remarks and interest. Unfortunately, this is only an assumption based on literary sources. We have added links to them in the text. We are going to do this in our future research.

Point 16: I would like to see the characteristic TLC plates in the form of a drawing. What are these flavonoids with Rf 0.83, 0.71, 0.78, 0.42, 0.17, etc. Is there this data?

Response 16: Thanks for the expert reminder and interest.

Generally, the authors also added supplementary material “AppendixA , Figure S2”, where can observe this.

Point 17: Line 453:PGPM, try to use one thing. Too free usage of abriveatures. Line 558: PGP try to use one thing. Too free usage of abriveatures. Line 523 ISR – abbreviation

Response 17: Thank you for your comments, the question regarding the abbreviation will be reviewed and corrected.

Point 18: The work was carried out (leaf samples were taken) in 2017-18, and funding was in 2021-22?

Response 18: Thank you for your comments, indeed, in 2017-2018, funding was provided from an additional project and at the expense of sponsorship assistance.

The authors corrected it in the text “grant number 110/540 (0117U004402) 2017-2019 "Biotechnological solutions of the gene pool preservation of historically valuable centuries-old trees".

Round 2

Reviewer 2 Report

I have no further comments. The authors replied to all queries/previous comments.

Author Response

Thank you for a detailed review of the article, for your comments and corrections.

We tried to correct and take everything into account. 

Reviewer 3 Report

The authors have eliminated most of the faults, only one remains (regarding Alternaria) and there are a number of comments.

             The abstract mentions the impact on Alternaria and Alternaria spp. is also mentioned as test objects (163-164, 393, 403), but they do not appear in the results section, there is also no information about them in additional materials (it is recommended to either remove one of two from the abstract and objects, or supplement the results and methods section).

 32: cinerea in italics

132: I think it should be pointed out that 200 years is an approximate age — for example, "about 200 years".

157: the degree is written in uppercase.

355: D. in italics.

             You mention bacteria of the genus Delftiatia, but it is probably Delftia.

Author Response

Dear Reviewer,

Thank you for a detailed review of the article, for your comments and corrections.

We tried to correct and take everything into account. In particular, the last wishes were also considered, such as:

Point 1:

The abstract mentions the impact on Alternaria and Alternaria spp. is also mentioned as test objects (163-164, 393, 403), but they do not appear in the results section, there is also no information about them in additional materials (it is recommended to either remove one of two from the abstract and objects, or supplement the results and methods section)?

Response 1: We mention Alternaria spp. as a part of the literature review. But we also corrected the abstract.

Point 2: Line  32: cinerea in italics.

Response 2: Thanks for the expert reminder. The authors corrected this.

Point 3: Line 132: I think it should be pointed out that 200 years is an approximate age — for example, "about 200 years".

Response 3: Thanks for the expert reminder. An apt remark, we also think that it is better to point out "about 200 years".

Point 4: Line 157: the degree is written in uppercase.

Response 4: Thanks for the expert reminder. The authors corrected this.

Point 5: Line 355: D. in italics.

Response 5: Thanks for the expert reminder. The authors corrected this.

Point 6: You mention bacteria of the genus Delftiatia, but it is probably Delftia.

Response 6: Thanks for the expert reminder. This is the authors mistake an it was corrected.
